# BraiNav: Incorporating Human Brain Activity to Enhance Robustness in Embodied Visual Navigation

## Abstract

Recent research shows that standard navigation agents significantly underperform and even fail in the presence of various visual corruptions. Unlike embodied agents, the human brain's visual system can robustly perceive the environment and extract the necessary information to complete the visual tasks. In this paper, we propose a two-phase **Brai**n-Machine integration **Nav**igation method called **BraiNav**, which incorporates neural representations derived from human brain activity to enhance robustness against visual corruptions. In the first phase, a brain encoder, built upon a recently advanced self-supervised pretrained model, is trained on a large-scale human brain activity dataset and then frozen for downstream visual navigation. In the second phase, neural representations harboring high-level cognitive information from the human brain are constructed based on the pretrained frozen brain encoder. Additionally, we propose a multimodal fusion method based on cross-attention to obtain more consistent brain-visual joint representations, which are then used to learn the navigation policy. Sufficient experiments demonstrate that the proposed method exhibits higher robustness against various visual corruptions compared to standard navigation agent and multiple computer vision-enhanced agents. Our study pioneers the incorporation of human brain activity into embodied AI, aiming to catalyze further cross-disciplinary collaboration with computational neuroscience.

## 1 Introduction

Embodied visual navigation (Anderson et al., 2018; Batra et al., 2020), one of the most researched topics in embodied AI (Duan et al., 2022), requires agents to make action decisions based on egocentric observations to achieve their goals. Despite the remarkable progress in embodied visual navigation, efforts (Wijmans et al., 2019; Zhao et al., 2021; Zhang et al., 2023) have primarily focused on training agents to generalize to unseen environments, assuming similarities between training and testing environments. However, a major challenge in this field is ensuring agent generalization across environments with different visual appearances (Chattopadhyay et al., 2021; Rajič, 2022).

Following standard protocol, agents are trained on a set of scenes and evaluated on unseen scenes, which entails two types of evaluation. The first type (Figure 1 (a)) assesses generalization performance on clean observations, where existing navigation methods have demonstrated high performance. The second type (Figure 1 (b)) introduces visual corruption, such as defocus blur, simulating real-life challenges. Assessing robustness to such corruption alongside generalization performance poses a greater challenge. While previous studies have utilized standard deep learning techniques, such as data augmentation and self-supervised adaptation (Chattopadhyay et al., 2021), to enhance robustness, there remains significant room for improvement in fully recovering lost navigation performance.

Unlike deep models, the human brain's visual system possesses superior capabilities in processing high-level semantic information from images, going beyond basic features like color, shape, and texture. As shown in Figure 1, humans can achieve comparable performance in navigation tasks under corrupted observations, indicating the robustness of the human visual system. Recent researches have demonstrated improved performance in deep learning models by incorporating neural repre-

sentations (Fong et al., 2018; Li et al., 2019; Nishida et al., 2020; Dapello et al., 2020; Fel et al., 2022; Liu et al., 2023; Shah et al., 2024). Notably, to the best of our knowledge, no prior study has utilized activity data collected from human brains as guidance for embodied visual navigation.

Motivated by these discussions, we investigated the potential of leveraging human brain activity for embodied visual navigation, proposing a two-phase **Brai**n-Machine integration **Nav**igation method, called **BraiNav**. **In the first phase**, a brain encoder model, built upon a recently advanced self-supervised pretrained model, is trained on a large-scale functional magnetic resonance imaging (fMRI) dataset. Following pretraining, the brain encoder is frozen for downstream visual navigation tasks. **In the second phase**, the navigation agent learns through deep reinforcement learning (DRL) (Sutton & Barto, 2018), comprising four major components: a brain module, a visual-target module, a fusion module, and a policy module. At each timestep, the agent obtains RGB observation and target localization. The RGB observation is initially processed by the brain module to extract neural representations, constructed based on the pretrained frozen brain encoder. Simultaneously, the visual-target module generates visual-target representations from RGB observation and tar-

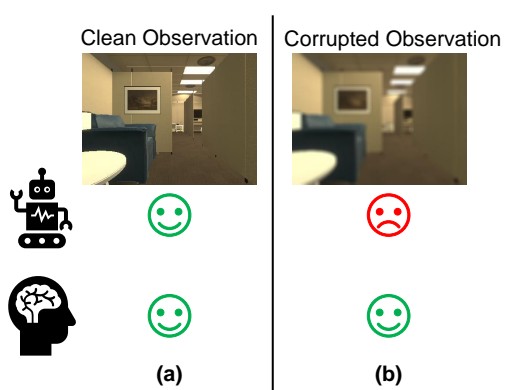

Figure 1: Comparison of human brain's visual system and embodied navigation agents in the presence of visual corruption. The human brain's visual system demonstrates superior robustness compared to navigation agents when observations are visually corrupted.

get localization. Then, a cross-attention (Vaswani et al., 2017) based multimodal fusion module is proposed to jointly learn the neural and visual-target representations to obtain the brain-visual joint representations. Finally, the joint representations are fed into the GRU-based (Cho et al., 2014) policy module, and the agent is trained using the Proximal Policy Optimization (PPO) algorithm (Schulman et al., 2017). Our results demonstrate that the BraiNav shows higher robustness against various visual corruptions compared to standard navigation agent and multiple computer vision-enhanced agents. To summarize, our main contributions are as follows:

- We introduce BraiNav, the first embodied navigation method incorporating human brain activity to enhance agent robustness against visual corruptions.

- We propose a multimodal fusion method based on cross-attention, yielding more consistent brain-visual joint representations.

- Sufficient experiments demonstrate that BraiNav exhibits superior robustness against various visual corruptions compared to standard navigation agent and multiple computer vision-enhanced agents. Our source code will be made available online post-publication.

## 2 METHODOLOGY

### 2.1 TASK DEFINITION

In this paper, we focus solely on PointNav. The navigation agent is trained using deep reinforcement learning. At each timestep $t$, the agent obtains the egocentric RGB observation $o_t$ and target location $l_t$. The agent then takes a predicted action to reach the target in as few timesteps as possible. There are four available actions for the agent: move forward (0.25m), turn left (30°), turn right (30°), and stop. An episode is considered successful if the agent stops within 0.2m of the target and executes a maximum of 300 steps. Unlike previous configurations where the RGB observation is clean during evaluation, in our task, the observation is subject to various visual corruptions to evaluate the robustness of the navigation agent.

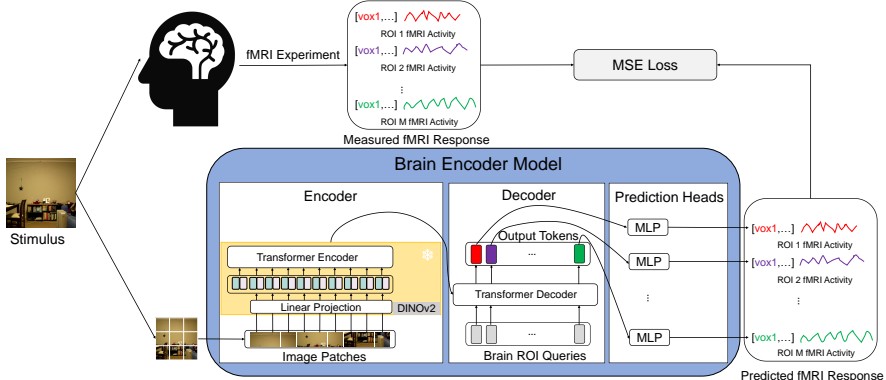

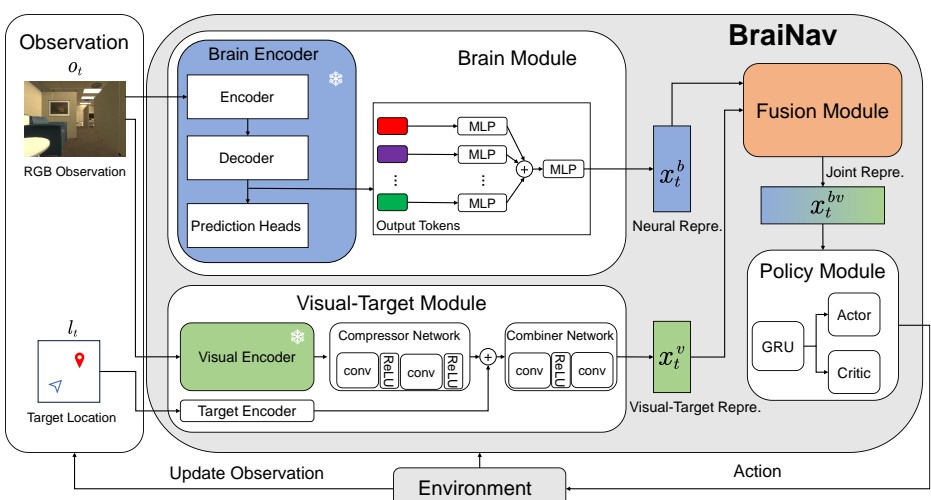

Figure 2: Overall Framework of BraiNav. **Phase 1:** The brain encoder model receives stimulus and outputs predicted fMRI responses, which are then supervised pretrained with experimentally measured fMRI responses. **Phase 2:** At each timestep, the RGB observation and target localization are fed into the brain module and visual-target module to obtain neural representation and visual-target representation, respectively. The multimodal fusion module combines these two different modality representations and outputs a consistent brain-visual joint representation, which is used to train the navigation agent.

## 2.2 OVERALL FRAMEWORK OF BRAINAV

BraiNav is designed with two sequential phases, as outlined in the Figure 2. In phase 1, a brain encoder model, built upon a self-supervised pretrained frozen model, is trained on a large-scale fMRI dataset. The learned brain encoder model will be frozen to guide the visual navigation process in the next phase. The architecture and training details of the brain encoder model will be provided in Section 2.3.

In phase 2, our agent consists of four major components: a brain module, a visual-target module, a fusion module, and a policy module. The brain module $E_b$ contains the pretrained frozen brain encoder, which is used to construct neural representations $x_t^b$ harboring high-level cognitive information from the human brain(to be discussed in Section 2.3), expressed as follows:

$$x_t^b = E_b\left(o_t\right).\tag{1}$$

The visual-target module is not elaborately designed, as the main purpose of this paper is to demonstrate the effectiveness of human brain activity for robust navigation. Specifically, the visual-target module $E_v$ comprises a frozen visual encoder, a target encoder, a compressor network, and a combiner network, with reference to (Chattopadhyay et al., 2021). The visual encoder is a frozen ResNet-18 (He et al., 2016) pretrained on ImageNet (Deng et al., 2009), which extracts visual features from images with a $512 \times 7 \times 7$ output. The compressor network consists of two convolutional layers, each followed by ReLU activation, transforming the output of the visual encoder to $512 \times 7 \times 7 \rightarrow 128 \times 7 \times 7 \rightarrow 32 \times 7 \times 7$. The target localization is a 2-dimensional polar coordinate $(r, \theta)$ mapped to a 32-dimensional target representation by the target encoder (a single-layer MLP), which is then expanded to $32 \times 7 \times 7$ dimensions. The outputs of the compressor network and target representation are concatenated and further processed by the combiner network, consisting of two convolutional layers that transform the output to $64 \times 7 \times 7 \rightarrow 128 \times 7 \times 7 \rightarrow 32 \times 7 \times 7$. Finally, the output of the combiner network is flattened to obtain a 1568-dimensional visual-target representation $x_t^v$, expressed as follows:

$$x_t^v = E_v\left(o_t, l_t\right). \tag{2}$$

The neural representation and visual-target representation belong to different modalities, and the intuitive $\mathrm{concatenate}$ method cannot yield promising result because inter-modal interactions are not fully exploited (Du et al., 2023; Chen et al., 2023). To address this problem, we propose a multimodal fusion module $E_f$ (to be discussed in Section 2.4) based on cross-attention (Vaswani et al., 2017) to obtain a more consistent brain-visual joint representation $x_t^{bv}$, expressed as follows:

$$x_t^{bv} = E_f\left(x_t^b, x_t^v\right). \tag{3}$$

The joint representation is fed into a GRU with 512 hidden units, along with the previous hidden state. The GRU outputs a 512-dimensional vector and the next hidden state, followed by two independent MLPs(marked as $\mathrm{Actor}$ and $\mathrm{Critic}$) that receive the 512-dimensional vector and output 4-dimensional action logits and a 1-dimensional scalar value, respectively. At timestep $t$, the reward received by the agent can be expressed as,

$$r_t = R_{success} - \triangle_t^{Geo} + \lambda, \tag{4}$$

where $R_{success} = 10$ denotes the reward obtained for a successful episode, $\triangle_t^{Geo}$ denotes the change in geodesic distance to the target at a single timestep interval, and $\lambda = -0.01$ denotes the movement penalty. The agent is trained using the Proximal Policy Optimization (PPO) (Schulman et al., 2017) algorithm.

## 2.3 BRAIN ENCODER PRETRAINING

The brain encoder model follows the method proposed in (Adeli et al., 2023) and consists of three components: an encoder, a decoder, and prediction heads, as shown in Figure 2. In the encoder, the input image is first divided into fixed 14 × 14 patches. These image patches are then processed by DINOv2 (Oquab et al., 2023), a recent advanced foundation model capable of extracting high-performance visual features. Specifically, the DINOv2 used is a self-supervised pretrained frozen ViT-B/14 (Dosovitskiy et al., 2020) model, which consists of a linear projection layer and a Transformer encoder. Each image patch is projected into a 768-dimensional patch embedding by the linear projection layer, and an additional learnable patch embedding is prepended to the sequence, denoted as CLS. Position embeddings are added to the patch embeddings before they are fed into the Transformer encoder, which consists of 12 stacked Transformer blocks (Vaswani et al., 2017).

The decoder is a single-layer Transformer with a feed-forward dimension of 2048 with 16 attention heads. The brain ROI queries consist of $M$ learnable positional encodings corresponding to different brain ROIs in each hemisphere. Specifically, 16 queries are used, with 8 for each hemisphere (7 streams add 1 for all the vertices labeled 'unknown'). The Transformer decoder utilizes the output from the encoder to transform these queries into output tokens.

The prediction heads comprise 16 MLPs that map the output tokens from the decoder to the fMRI responses of the corresponding ROIs. Each output token is mapped to a vector by the MLP with the same number of voxels as the corresponding hemisphere (19,004 for the left hemisphere and 20,544 for the right hemisphere), with voxels not belonging to that ROI set to 0 using a mask. The

prediction loss is computed by calculating the Mean Square Error (MSE) between the predicted and measured fMRI responses voxel by voxel.

After pretraining, we construct neural representations for downstream visual navigation using the pretrained frozen brain encoder, as shown in the brain module in Figure 2. Specifically, we use the low-dimensional and noise-reduced output tokens from the brain encoder to construct neural representations, effectively exploiting the high-level information embedded in human brain activities. Each 768-dimensional output token from each ROI of the anatomical streams is mapped to 48 dimensions using an MLP, and the resulting vectors are concatenated and fed into another MLP to obtain 768-dimensional neural representation.

## 2.4 BRAIN-VISUAL MULTIMODAL LEARNING

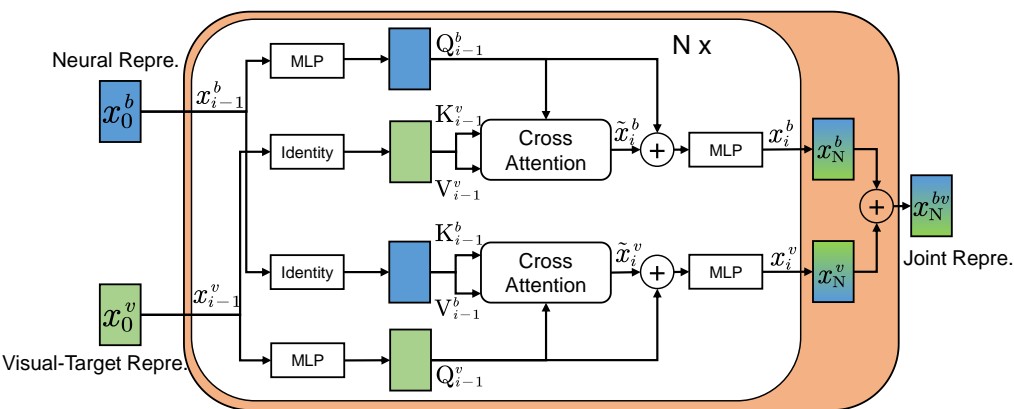

Figure 3: Architecture of the multimodal fusion module. The module is primarily composed of N stacked computational blocks, each consisting of two branches. Each branch processes neural representations and visual-target representations of different dimension, which are effectively fused at the end using cross-attention mechanisms.

The architecture of the multimodal fusion module is shown in Figure 3, consisting of N stacked computational blocks. Each computational block includes multiple Cross-Attention (CA) layers and MLPs, and contains two branches.

For simplicity, the timestep $t$ is omitted. In the brain branch (top), the Key embedding $\mathrm{K}_{i-1}^v \in \mathrm{R}^{\mathrm{V_{dim}}}$ and Value embedding $\mathrm{V}_{i-1}^v \in \mathrm{R}^{\mathrm{V_{dim}}}$ are first obtained by identity mapping the visual-target representation $x_{i-1}^v \in \mathrm{R}^{\mathrm{V_{dim}}}$,

$$\mathrm{K}_{i-1}^v, \mathrm{V}_{i-1}^v = \mathrm{Identity}\left(x_{i-1}^v\right). \tag{5}$$

To align the dimensions, the Query embedding $\mathrm{Q}_{i-1}^b \in \mathrm{R}^{\mathrm{V_{dim}}}$ is obtained by applying a singer-layer MLP to the neural representation $x_{i-1}^b \in \mathrm{R}^{\mathrm{B_{dim}}}$,

$$\mathrm{Q}_{i-1}^b = \mathrm{MLP}\left(x_{i-1}^b\right). \tag{6}$$

Then, the brain Query interacts with the visual-target Key and Value through CA, mathematically expressed as:

$$\tilde{x}_i^b = \mathrm{CA}\left(\mathrm{Q}^b, \mathrm{K}^v, \mathrm{V}^v\right)$$
$$= \mathrm{softmax}\left(\mathbf{q}\mathbf{k}^{\mathrm{T}}/\sqrt{\mathrm{B_{dim}/V_{dim}}}\right)\mathbf{v}, \tag{7}$$
$$\mathbf{q} = \mathrm{Q}^b\mathbf{W}_q, \mathbf{k} = \mathrm{K}^v\mathbf{W}_k, \mathbf{v} = \mathrm{V}^v\mathbf{W}_v,$$

where $\mathbf{W}_q$, $\mathbf{W}_k$, and $\mathbf{W}_q$ are learnable parameters. Finally, to align the dimensions, $x_i^b \in \mathrm{R}^{\mathrm{B_{dim}}}$ is obtained by mapping the output of CA with a residual shortcut through a single-layer MLP,

$$x_i^b = \mathrm{MLP}\left(\tilde{x}_i^b + \mathrm{Q}_{i-1}^b\right). \tag{8}$$

For the visual-target branch (bottom), the same procedure as the brain branch is performed, simply swapping Query with Key and Value. After N computational blocks, the final neural representation $x_N^b$ and visual-target representation $x_N^v$ are concatenated to produce the joint representation $x_N^{bv}$. Experimentally, the fusion module uses only one computational block (N = 1) for convergence considerations, where the cross-attention is a Transformer model with 3 heads.

## 3 EXPERIMENTS

### 3.1 DATASET AND ENVIRONMENT

The brain encoder is trained using the Natural Scenes Dataset (NSD) dataset (Allen et al., 2022), which is currently the largest and richest neuroimaging dataset. It provides high-quality whole-brain 7T fMRI responses from 8 subjects viewing $\sim$73,000 different natural scenes while performing a continuous recognition task. The color image stimuli viewed by the subjects come from the Common Objects in Context (COCO) dataset (Lin et al., 2014), which is richly annotated and widely used in computer vision.

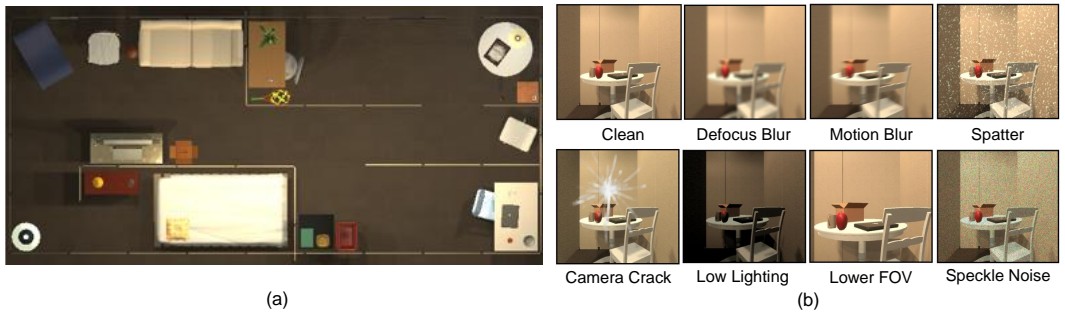

Figure 4: Navigation scene and visual corruptions. (a) An example of an indoor scene from RoboTHOR dataset, with each scene sized at 8.8m $\times$ 3.9m. (b) The top-left shows a clean RGB observation, while the remaining images display its corresponding corrupted observations. The illustrations of the corruptions are adapted from (Chattopadhyay et al., 2021).

The visual navigation agent will be trained and evaluated on the ROBUSTNAV benchmark (Chattopadhyay et al., 2021), built on top of the AI2-THOR simulator (Kolve et al., 2017) and the RoboTHOR dataset (Deitke et al., 2020). RoboTHOR contains 75 indoor scenes with different layouts(Figure 4(a) shows an example scene), with 60 used for training and 15 for evaluation. BraiNav is evaluated on 7 visual corruptions provided in ROBUSTNAV (shown in the Figure 4(b)): Defocus Blur, Motion Blur, Spatter, Camera Crack, Low Lighting, Lower FOV, and Speckle Noise. A detailed description of these visual corruptions can be found in (Chattopadhyay et al., 2021).

### 3.2 EXPERIMENTS CONFIGURATION

For brain encoder pretraining, the learning rate is set to 1e-4, the batch size to 32, the number of training epochs to 20, the gradient clip to 0.1, the weight decay to 1e-4, and dropout to 0.1. The subsequent brain encoder is trained using the fMRI data from subject 01, with results from other subjects presented shown in Appendix B.1. Additionally, the original 425$\times$425 resolution images are resized to 224$\times$224 to match the observation image resolution for downstream visual navigation tasks.

For BraiNav training, the learning rate is set to 3e-4, the discount factor to 0.99, the rollout length to 128, and the total training frames to 75M. The corruption level is consistently set to 5, as in ROBUSTNAV, where level 5 indicates the most severe corruptions. Training is conducted on a single NVIDIA GeForce RTX 4090 GPU.

### 3.3 EVALUATION METRICS

We use two common metrics in visual navigation: Success Rate (SR) and Success weighted by Path Length(SPL). SR indicates the percentage of successful episodes, defined as follows:

$$\text{SR} = \frac{1}{\text{K}} \sum_{i=1}^{\text{K}} \Pi_i, \tag{9}$$

Where K is the number of episodes in the evaluation and $\Pi_i$ is the binary indicator whether the $i$-th episode is successful (1 if successful, otherwise 0). SPL is the percentage of the path length of successful episodes to the shortest path, defined as follows:

$$\text{SPL} = \frac{1}{\text{K}} \sum_{i=1}^{\text{K}} \Pi_i \frac{l_i}{\max(p_i, l_i)}, \tag{10}$$

where $l$ is the shortest path length and $p$ is the agent's path length. Higher SR and SPL indicate that the navigation agent is more effective and efficient.

### 3.4 PERFORMANCE OF BRAIN ENCODER

After pretraining, we evaluate the prediction accuracy of the brain encoder on the ROIs used to construct the neural representations. We compute the noise ceiling for each vertex using response data from the subjects' three trials of the same stimulus image, and then average these values across each ROI. Details of the calculation methodology and results are provided in Appendix B.2. Next, we evaluate the brain encoder's performance by calculating the *Pearson correlation coefficient* between its predicted fMRI responses and the actual values. To determine the noise-normalized prediction accuracy, we divide the encoder's prediction accuracy by the corresponding noise ceiling, as summarized in Table 1.

The results indicate that the brain encoder effectively predicts the fMRI response to visual stimuli. High prediction accuracy ensures that the neural representations used for learning the navigation policy contain cognitive processing information from the human brain.

### 3.5 COMPARISON WITH STANDARD NAVIGATION AGENT

We compare the proposed method with the standard navigation agent (Chattopadhyay et al., 2021), as shown in Table 2. BraiNav outperforms the standard navigation agent across 6 visual corruptions, achieving varying degrees of performance improvement. Notably, on Speckle Noise and Defocus Blur, BraiNav shows significant improvements, with absolute improvements of (15.77%SPL, 17.75%SR) and (11.89%SPL, 11.37%SR), respectively. For other corruptions, the absolute improvements are (4.93%SPL, 1.37%SR) for Motion Blur, (2.76%SPL, 3.18%SR) for Spatter, (1.8%SPL, 0.73%SR) for Low Lighting, and (0.83%SPL, 0.46%SR) for Lower FOV.

It is worth noting that our method exhibits remarkable performance improvement on Defocus Blur. Neuroscience studies (Mon-Williams et al., 1998; Webster et al., 2002; Zhu et al., 2013) suggest the existence of a perceptual mechanism in the human brain that regulates image defocus. In conclusion, these experimental results demonstrate that our proposed navigation method, BraiNav, surpasses the standard navigation agent across various visual corruptions.

### 3.6 COMPARISON WITH COMPUTER VISION-ENHANCED AGENTS

We also compare our proposed method with several computer vision-enhanced agents outlined in (Chattopadhyay et al., 2021), as shown in Table 2.

**Standard Agent+AP**: Standard Agent introduces an auxiliary action prediction task.

**Standard Agent+AP+SS-Adapt**: Standard Agent+AP introduces self-supervised adaptation on specific corruptions.

**Standard Agent+RP**: Standard Agent introduces an auxiliary rotation prediction task.

Table 1: Noise-normalized prediction accuracy of the brain encoder.

| Hemisphere | ROI | | | | | | |
|---|---|---|---|---|---|---|---|
| | early | midventral | midlateral | midparietal | ventral | lateral | parietal |
| Left Hemisphere | 0.7358 | 0.8254 | 0.8116 | 0.8432 | 0.7239 | 0.9179 | 0.7572 |
| Right Hemishpere | 0.7144 | 0.7910 | 0.7950 | 0.7239 | 0.8213 | 0.8761 | 0.6816 |

Table 2: Comparison with standard and computer vision-enhanced navigation agents. We compare BraiNav with six approaches proposed in (Chattopadhyay et al., 2021). The results are highlighted with best and second .

| Approach | Visual Corruption | | | | | | | | | | | | | | | |
|---|---|---|---|---|---|---|---|---|---|---|---|---|---|---|---|---|
| | Clean | | Spatter | | Speckle Noise | | Camera Crack | | Lower FOV | | Defocus Blur | | Motion Blur | | Low Lighting | |
| | SR | SPL | SR | SPL | SR | SPL | SR | SPL | SR | SPL | SR | SPL | SR | SPL | SR | SPL |
| Standard Agent | 98.82 | 83.13 | 33.58 | 24.72 | 67.42 | 48.57 | 82.07 | 63.83 | 42.49 | 31.73 | 75.89 | 53.55 | 95.72 | 73.37 | 94.36 | 75.15 |
| Standard Agent +AP | 98.45 | 83.28 | 20.38 | 15.70 | 65.61 | 47.03 | 72.70 | 56.82 | 45.68 | 35.14 | 83.35 | 61.51 | 94.81 | 74.30 | 92.17 | 76.11 |
| Standard Agent +AP+SS-Adapt | 37.31 | 31.03 | 14.19 | 10.29 | \ | \ | 57.87 | 46.72 | 32.94 | 26.09 | 40.95 | 33.35 | \ | \ | \ | \ |
| Standard Agent +RP | 98.73 | 82.53 | 23.48 | 18.63 | 78.98 | 55.92 | 67.06 | 53.70 | 44.95 | 32.74 | 32.21 | 22.47 | 91.63 | 65.27 | 89.81 | 67.38 |
| Standard Agent +RP+SS-Adapt | 94.63 | 77.25 | 61.06 | 47.16 | \ | \ | 60.42 | 49.37 | 50.59 | 36.10 | 79.16 | 62.74 | \ | \ | \ | \ |
| Standard Agent +Data Aug | 98.45 | 81.08 | 23.93 | 18.41 | 77.25 | 57.95 | 88.44 | 71.57 | 71.70 | 54.54 | 81.26 | 61.32 | 96.91 | 75.97 | 97.27 | 78.74 |
| **BraiNav** | 97.73 | 80.88 | 36.76 | 27.48 | 85.17 | 64.34 | 77.80 | 60.25 | 42.95 | 32.56 | 87.26 | 65.44 | 97.09 | 78.30 | 95.09 | 76.95 |

**Standard Agent+RP+SS-Adapt**: Standard Agent+RP introduces self-supervised adaptation on specific corruptions.

**Standard Agent+Data Aug**: Standard Agent introduces various data augmentation during training.

BraiNav outperforms all the aforementioned computer vision-enhanced agents across 3 visual corruptions. Specifically, the absolute improvements are (6.39%SPL, 6.19%SR) for Speckle Noise, (2.7%SPL, 3.91%SR) for Defocus Blur, and (2.33%SPL, 0.18%SR) for Motion Blur. Additionally, BraiNav achieves the second best results on Spatter and Low Lighting, demonstrating competitive performance on the remaining visual corruptions. Additional experiments and detailed results of the comparison methods are provided in the Appendix B.3 and Appendix B.4.

### 3.7 ABLATION STUDY

BraiNav consists of two key components: the neural representation from the pretrained brain encoder and the multimodal fusion module. In this section, we first analyze the impact of the neural representation, followed by an evaluation of the multimodal fusion module's effectiveness.

**For the first ablation experiment**, we concatenate the original DINOv2 CLS representation with the visual-target representation to form the joint representation, labeled as **BraiNav w/o NR** in Table 3.

Except for Lower FOV, BraiNav consistently outperforms BraiNav w/o NR across clean and the other 6 visual corruptions, achieving performance improvements to varying degrees. Specifically, the absolute improvements are (14.6%SPL, 18.56%SR) for Speckle Noise, (9.08%SPL, 13.01%SR) for Spatter, (7.42%SPL, 9.83%SR) for Low Lighting, (3.84%SPL, 5.37%SR) for Motion Blur, (1.9%SPL, 8.64%SR) for Defocus Blur, and (2.55%SR) for Camera Crack. These results highlight that neural representation derived from human brain activity significantly enhances BraiNav's robustness against visual corruptions, beyond the deep representation from DINOv2.

**For the second ablation experiment**, we replace the multimodal fusion module in BraiNav with the concatenate method, and the experimental results are labeled **BraiNav w/o MF** as shown in Table 3.

Table 3: Contribution of each component. **BraiNav w/o NR** presents experimental results without the neural representations (NR) derived from human brain activity. **BraiNav w/o MF** presents experimental results excluding the multimodal fusion (MF) module.

| Approach | Visual Corruption | | | | | | | | | | | | | | | |
|---|---|---|---|---|---|---|---|---|---|---|---|---|---|---|---|---|
| | Clean | | Spatter | | Speckle Noise | | Camera Crack | | Lower FOV | | Defocus Blur | | Motion Blur | | Low Lighting | |
| | SR | SPL | SR | SPL | SR | SPL | SR | SPL | SR | SPL | SR | SPL | SR | SPL | SR | SPL |
| BraiNav w/o NR | 95.45 | 79.74 | 23.75 | 18.40 | 66.61 | 49.74 | 75.25 | 61.40 | 48.95 | 38.63 | 78.62 | 63.54 | 91.72 | 74.46 | 85.26 | 69.53 |
| BraiNav w/o MF | 99.18 | 83.14 | 23.38 | 18.38 | 54.69 | 42.54 | 79.25 | 62.36 | 38.71 | 30.43 | 76.34 | 53.45 | 92.54 | 71.96 | 91.63 | 72.52 |
| BraiNav | 97.73 | 80.88 | 36.76 | 27.48 | 85.17 | 64.34 | 77.80 | 60.25 | 42.95 | 32.56 | 87.26 | 65.44 | 97.09 | 78.30 | 95.09 | 76.95 |

Except for Camera Crack, the multimodal fusion module enhances BraiNav's robustness across the other 6 visual corruptions. Specifically, the absolute improvements are (21.8%SPL, 30.48%SR) for Speckle Noise, (11.99%SPL, 10.92%SR) for Defocus Blur, (9.1%SPL, 13.38%SR) for Spatter, (6.34%SPL, 4.55%SR) for Motion Blur, (4.43%SPL, 3.46%SR) for Low Lighting, and (2.13%SPL, 4.24%SR) for Lower FOV. Overall, these findings demonstrate that the multimodal fusion module further enhances the robustness of BraiNav. Additional ablation experiments and detailed results are provided in the Appendix B.5.

## 4 DISCUSSION AND CONCLUSION

In this paper, we introduce BraiNav, a novel framework designed to address the robustness challenges in embodied visual navigation. Our two-phase method leverages human brain activity to enhance the navigation agent's resilience to visual corruption. In the first phase, we pretrain a brain encoder model with DINOv2 as the backbone on a large-scale fMRI dataset. In the second phase, we utilize the pretrained frozen brain encoder to construct neural representations that encapsulate high-level cognitive information from the human brain. Additionally, we develop a multimodal fusion module based on cross-attention to facilitate the learning of consistent brain-visual joint representations for navigation policy acquisition. We evaluate BraiNav's navigation performance across multiple visual corruptions, demonstrating its superior robustness compared to standard visual navigation agent and multiple computer vision-enhanced agents. More importantly, our research bridges embodied AI and neuroscience, showcasing the potential for translating insights from neuroscience into advancements in embodied AI.

To create a brain-like representation for the navigation agent, BraiNav first employs a brain encoder pretrained on fMRI data to obtain neural representations, followed by a multimodal fusion module to achieve more consistent joint representations. Thus, enhancements in both the brain encoder and the multimodal fusion module are expected to yield better performance. Future research could explore different brain encoder architectures (Yang et al., 2024) and multimodal fusion methods (Mao et al., 2023). Furthermore, the brain module in BraiNav is decoupled from the specific task; it can receive images and output brain-like representations to improve the agent's robustness to visual corruptions for other embodied tasks. Future studies could apply BraiNav's brain-like representations to a broader range of embodied tasks (Wan et al., 2023).

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

# A    RELATED WORK

## A.1    EMBODIED VISUAL NAVIGATION

Two widely studied tasks in embodied visual navigation are PointGoal Navigation (PointNav) (Anderson et al., 2018) and ObjectGoal Navigation (ObjectNav) (Batra et al., 2020). PointNav involves navigating to a specified goal coordinate in a global reference frame, while ObjectNav requires the agent to find an instance of a specified object. Thanks to high-quality simulators (Kolve et al., 2017; Savva et al., 2019) and datasets (Chang et al., 2017; Deitke et al., 2020), significant advancements has been made in embodied visual navigation (Wijmans et al., 2019; Zhao et al., 2021; Partsey et al., 2022; Zhang et al., 2023), especially in the PointNav task, which is often considered "solved" (Wijmans et al., 2019).

However, existing studies have largely overlooked the robustness of agents, a critical aspect for real-world applications. To address this gap, (Chattopadhyay et al., 2021) proposed ROBUSTNAV, a framework for analyzing the robustness of navigation agents. ROBUSTNAV quantifies the performance of embodied navigation agents under various common visual and dynamic corruptions. Extensive experiments demonstrated that navigation agents trained in simulation often exhibit significant performance drops when evaluated in corrupted environments. Furthermore, (Rajič, 2022) conducted a robustness analysis of two successful agents from the 2021 Habitat Challenge, revealing varying degrees of performance deterioration in corrupted settings. (Lee et al., 2022) proposed a self-supervised domain adaptation method with map style transfer to boost agent robustness against visual and dynamic perturbations. (Piriyajitakonkij et al., 2024) introduced TTA-Nav, which enhances navigation performance across visual corruptions using a top-down decoder. Our approach differs from theirs in that we exploits the capabilities of the human brain visual system by introducing brain-like representations for agents to enhance robustness.

## A.2    DEEP LEARNING AND BRAIN ACTIVITY INTEGRATION

Recent studies have explored techniques to enhance computer vision models by integrating neural network features with human brain activity. (Fong et al., 2018) improved image classification in Convolutional Neural Networks (CNNs) by incorporating voxel responses from fMRI during training, making the decision surface of the classifier more consistent with brain representations. (Li et al., 2019) regularized CNNs using cortical representations from neuroscience data, enhancing robustness against adversarial attacks. (Nishida et al., 2020) introduced a brain-mediated transfer learning (TL) method, transforming the feature representation of audiovisual input in CNNs into brain representations, achieving superior performance in estimating human cognitive and behavioral labels compared to standard TL. Additionally, (Dapello et al., 2020) developed VOneNet to simulate the primary visual cortex, thereby enhancing CNN classification robustness. (Fel et al., 2022) presented a neural harmonizer that aligns deep neural networks with human visual strategies, resulting in improved classification accuracy. More recently, (Liu et al., 2023) proposed a brain-machine coupled learning method that utilized visual images and electroencephalogram (EEG) signals for training models in facial emotion recognition, demonstrating improved generalization. (Shah et al., 2024) developed an image transform that simulates peripheral vision, boosting DNN robustness against adversarial attacks. These studies indicate that combining neural network features with human brain activity can yield improvements. However, all of these works focus on image classification tasks, while our approach tackles the more complex domain of embodied visual navigation, a topic that has not been extensively explored. Furthermore, we leverage a much larger and higher-quality fMRI dataset.

## A.3    BRAIN ENCODING MODEL

Brain encoding models are capable of predicting fMRI data from humans viewing visual stimuli, which is crucial for understanding how information is represented in the brain (Naselaris et al., 2011; Wen et al., 2018). (Kay et al., 2008) developed a linear encoding model based on a Gabor wavelet pyramid to predict brain responses to stimulus images. (Khosla et al., 2020) proposed an encoding model that incorporates visual attention, resulting in significant improvements in predicting neural responses. More recent studies (Adeli et al., 2023; Yang et al., 2024) have utilized

Table 4: Noise-normalized prediction accuracy of the brain encoder for subject 02.

| Hemisphere | ROI | | | | | | |
|---|---|---|---|---|---|---|---|
| | early | midventral | midlateral | midparietal | ventral | lateral | parietal |
| Left Hemisphere | 0.7356 | 0.8213 | 0.8692 | 0.6995 | 0.9106 | 0.8831 | 0.7074 |
| Right Hemishpere | 0.7491 | 0.8166 | 0.8250 | 0.7682 | 0.8588 | 0.9891 | 0.7396 |

Table 5: Comparison with standard and computer vision-enhanced navigation agents for subject 02. We compare BraiNav with six approaches proposed in (Chattopadhyay et al., 2021). The results are highlighted with best and second.

| Approach | Visual Corruption | | | | | | | | | | | | | | | |
|---|---|---|---|---|---|---|---|---|---|---|---|---|---|---|---|---|
| | Clean | | Spatter | | Speckle Noise | | Camera Crack | | Lower FOV | | Defocus Blur | | Motion Blur | | Low Lighting | |
| | SR | SPL | SR | SPL | SR | SPL | SR | SPL | SR | SPL | SR | SPL | SR | SPL | SR | SPL |
| Standard Agent | 98.82 | 83.13 | 33.58 | 24.72 | 67.42 | 48.57 | 82.07 | 63.83 | 42.49 | 31.73 | 75.89 | 53.55 | 95.72 | 73.37 | 94.36 | 75.15 |
| Standard Agent +AP | 98.45 | 83.28 | 20.38 | 15.70 | 65.61 | 47.03 | 72.70 | 56.82 | 45.68 | 35.14 | 83.35 | 61.51 | 94.81 | 74.30 | 92.17 | 76.11 |
| Standard Agent +AP+SS-Adapt | 37.31 | 31.03 | 14.19 | 10.29 | \ | \ | 57.87 | 46.72 | 32.94 | 26.09 | 40.95 | 33.35 | \ | \ | \ | \ |
| Standard Agent +RP | 98.73 | 82.53 | 23.48 | 18.63 | 78.98 | 55.92 | 67.06 | 53.70 | 44.95 | 32.74 | 32.21 | 22.47 | 91.63 | 65.27 | 89.81 | 67.38 |
| Standard Agent +RP+SS-Adapt | 94.63 | 77.25 | 61.06 | 47.16 | \ | \ | 60.42 | 49.37 | 50.59 | 36.10 | 79.16 | 62.74 | \ | \ | \ | \ |
| Standard Agent +Data Aug | 98.45 | 81.08 | 23.93 | 18.41 | 77.25 | 57.95 | 88.44 | 71.57 | 71.70 | 54.54 | 81.26 | 61.32 | 96.91 | 75.97 | 97.27 | 78.74 |
| **BraiNav** | 99.36 | 84.14 | 39.03 | 28.36 | 80.78 | 60.92 | 80.98 | 62.18 | 48.86 | 37.26 | 85.71 | 63.70 | 94.63 | 73.95 | 94.38 | 75.47 |

self-supervised pretrained Visual Transformer (ViT) (Dosovitskiy et al., 2020) models as backbones to extract features from stimulus images, achieving excellent performance.

# B MORE EXPERIMENTS

## B.1 DIFFERENT SUBJECT

We also train the brain encoder using the fMRI data from subj 02 and utilized the pretrained frozen brain encoder for downstream navigation tasks. We compute the noise ceiling for each vertex using response data from the subjects' three trials and then average these values across each ROI. Details of the calculation methodology and results are provided in Appendix B.2. Next, we evaluate the brain encoder's performance by calculating the *Pearson correlation coefficient* between its predicted fMRI responses and the actual values. To determine the noise-normalized prediction accuracy, we divide the encoder's prediction accuracy by the corresponding noise ceiling, as summarized in Table 4. The results indicate that the brain encoder effectively predicts the fMRI responses to visual stimuli. High prediction accuracy ensures that the neural representations used for navigation policy learning incorporate relevant cognitive processing information from the human brain.

Next, we compare the proposed method with standard and computer vision-enhanced agents, as shown in Table 5. BraiNav outperforms the standard navigation agent across clean and six visual corruptions, achieving varying degrees of performance improvements. Notably, for Speckle Noise and Defocus Blur, BraiNav achieves significant improvements, with absolute improvements of (12.35%SPL, 13.36%SR) and (10.15%SPL, 9.82%SR), respectively. For other corruptions, the absolute improvements are (5.53%SPL, 6.37%SR) for Lower FOV, (3.64%SPL, 5.45%SR) for Spatter, (0.58%SPL) for Motion Blur, and (0.32%SPL, 0.02%SR) for Low Lighting. Additionally, BraiNav outperforms all computer vision-enhanced agents on clean and 2 visual corruptions. In detail, the absolute improvements are (2.97%SPL, 1.8%SR) for Speckle Noise and (0.96%SPL, 2.36%SR) for Defocus Blur. Additionally, BraiNav achieves the second best results on Spatter, Lower FOV, and Low Lighting, demonstrating competitive performance on the remaining visual corruptions.

Table 6: Noise ceiling for subject 01.

| Hemisphere | ROI | | | | | | |
|---|---|---|---|---|---|---|---|
| | early | midventral | midlateral | midparietal | ventral | lateral | parietal |
| Left Hemisphere | 0.5965 | 0.5967 | 0.5971 | 0.5942 | 0.5857 | 0.5881 | 0.5870 |
| Right Hemishpere | 0.5945 | 0.5967 | 0.5888 | 0.5910 | 0.5863 | 0.5870 | 0.5873 |

Table 7: Noise ceiling for subject 02.

| Hemisphere | ROI | | | | | | |
|---|---|---|---|---|---|---|---|
| | early | midventral | midlateral | midparietal | ventral | lateral | parietal |
| Left Hemisphere | 0.6006 | 0.6056 | 0.5992 | 0.6047 | 0.5905 | 0.5921 | 0.5980 |
| Right Hemishpere | 0.5991 | 0.5983 | 0.5950 | 0.6013 | 0.5908 | 0.5941 | 0.5984 |

## B.2 NOISE CEILING ACROSS DIFFERENT SUBJECTS

For each stimulus image, subjects view it three times, resulting in 3-trial fMRI responses. For stimulus image $i$, let the three response values for vertex $j$ be denoted as $\text{rep}0_j^i, \text{rep}1_j^i, \text{rep}2_j^i$. Across $N$ stimulus images, vertex $j$ accumulates three sets of trial responses: $\text{rep}0_j = \left[\text{rep}0_j^1, ..., \text{rep}0_j^N\right]$, $\text{rep}1_j = \left[\text{rep}1_j^1, ..., \text{rep}0_j^N\right]$, $\text{rep}2_j = \left[\text{rep}2_j^1, ..., \text{rep}2_j^N\right]$. The average of these 3-trial responses is denoted as $\text{repm}_j$. The noise ceiling for vertex $j$ is then calculated as:

$$\text{nc}_j = \frac{\text{pearsonr}\left(\text{rep}0_j, \text{repm}_j\right) + \text{pearsonr}\left(\text{rep}1_j, \text{repm}_j\right) + \text{pearsonr}\left(\text{rep}2_j, \text{repm}_j\right)}{3}, \quad (11)$$

where pearsonr represents the Pearson correlation coefficient operator. After computing the noise ceilings for all vertices, we average them by ROI. The results are presented in Table 6 for subject 01 and Table 7 for subject 02.

## B.3 MORE COMPUTER VISION-ENHANCED AGENT

We develop a computer vision-enhanced agent leveraging the advanced self-supervised model, Masked Autoencoder (MAE) (He et al., 2022). The agent is first deployed in RoboTHOR indoor scenes to freely explore and collect 60,000 egocentric images as training data for MAE. These images are then masked and reconstructed for self-supervised pretraining. Specifically, the MAE encoder is implemented as a 12-layer Vision Transformer (ViT), while the decoder consists of an 8-layer ViT, with a masking ratio of 0.75. After pretraining, the encoder is retained as the agent's visual backbone, and the decoder is discarded. We compare the performance of our proposed BraiNav agent against the MAE-based agent under seven visual corruption scenarios. As shown in Table 8, BraiNav consistently outperforms the MAE-based agent across all corruption types.

## B.4 COMPARISON WITH AGENT BASED ON BRAIN-LIKE REPRESENTATIONS

We have replaced the brain encoder with CORnet-S (Kubilius et al., 2019), a compact, recurrent artificial neural network model that aligns closely with the anatomical structure and dynamic responses of the primate ventral visual stream. CORnet-S not only achieves high biological fidelity

Table 8: Comparison with MAE agent.

| Approach | Visual Corruption | | | | | | | | | | | | | |
|---|---|---|---|---|---|---|---|---|---|---|---|---|---|---|
| | Clean | | Spatter | | Speckle Noise | | Camera Crack | | Lower FOV | | Defocus Blur | | Motion Blur | | Low Lighting |
| | SR | SPL | SR | SPL | SR | SPL | SR | SPL | SR | SPL | SR | SPL | SR | SPL | SR | SPL |
| MAE Agent | 98.54 | 83.31 | 8.55 | 5.92 | 10.10 | 7.66 | 48.95 | 37.67 | 31.57 | 23.09 | 67.42 | 51.95 | 79.34 | 60.14 | 36.12 | 28.27 |
| BraiNav | 97.73 | 80.88 | 36.76 | 27.48 | 85.17 | 64.34 | 77.80 | 60.25 | 42.95 | 32.56 | 87.26 | 65.44 | 96.90 | 78.30 | 95.09 | 75.95 |

Table 9: Comparison with CORnet agent.

| Approach | Visual Corruption | | | | | | | | | | | | | | | |
|---|---|---|---|---|---|---|---|---|---|---|---|---|---|---|---|---|
| | Clean | | Spatter | | Speckle Noise | | Camera Crack | | Lower FOV | | Defocus Blur | | Motion Blur | | Low Lighting | |
| | SR | SPL | SR | SPL | SR | SPL | SR | SPL | SR | SPL | SR | SPL | SR | SPL | SR | SPL |
| CORnet Agent | 97.27 | 81.63 | 30.30 | 23.28 | 78.07 | 61.53 | 77.80 | 61.45 | 61.60 | 48.17 | 75.98 | 57.16 | 94.18 | 73.74 | 87.99 | 70.08 |
| BraiNav | 97.73 | 80.88 | 36.76 | 27.48 | 85.17 | 64.34 | 77.80 | 60.25 | 42.95 | 32.56 | 87.26 | 65.44 | 96.90 | 78.30 | 95.09 | 75.95 |

Table 10: Impact of different ROIs.

| Approach | Visual Corruption | | | | | | | | | | | | | | | |
|---|---|---|---|---|---|---|---|---|---|---|---|---|---|---|---|---|
| | Clean | | Spatter | | Speckle Noise | | Camera Crack | | Lower FOV | | Defocus Blur | | Motion Blur | | Low Lighting | |
| | SR | SPL | SR | SPL | SR | SPL | SR | SPL | SR | SPL | SR | SPL | SR | SPL | SR | SPL |
| Low-Level ROI | 96.54 | 79.71 | 34.03 | 25.17 | 78.14 | 59.57 | 77.43 | 61.09 | 44.59 | 34.35 | 73.97 | 55.30 | 90.72 | 72.45 | 94.44 | 72.73 |
| High-Level ROI | 97.27 | 80.53 | 37.03 | 27.61 | 87.52 | 65.81 | 79.25 | 63.05 | 44.31 | 34.08 | 86.90 | 66.03 | 96.45 | 78.52 | 96.81 | 75.82 |

but also excels in both neuroscience and machine learning benchmarks. CORnet-S extracts visual features from images with a 1000-dimensional brain-like representations, followed by a linear layer converted to 768 dimensions. The results are presented in Table 9. BraiNav outperforms the CORnet Agent on Spatter, Speckle Noise, Defocus Blur, Motion Blur, and Low Lighting. Conversely, the CORnet Agent demonstrates strong performance on Camera Crack and Lower FOV. These results highlight that brain-like representations can enhance the robustness of embodied navigation agent.

## B.5 IMPACT OF DIFFERENT ROIS

Early brain regions are primarily responsible for processing low-level visual information, while ventral, lateral, and parietal regions handle high-level visual information (Allen et al., 2022). To investigate the contribution of features from different brain regions, we categorize the brain into two groups: low-level regions (early) and high-level regions (ventral, lateral, and parietal). Neural representations for visual navigation are then constructed using output tokens from each group. The results are presented in Table 10.

Except for the Lower FOV corruption, representations derived from high-level brain regions significantly outperform those from low-level regions across all other corruptions, demonstrating strong consistency. These findings further confirm that the brain encoder effectively generates neural representations containing cognitive information from the human brain. Moreover, representations from higher brain regions exhibit greater robustness.

