# OpenReview forum: "BraiNav: Incorporating Human Brain Activity to Enhance Robustness in Embodied Visual Navigation"
_ICLR.cc/2025/Conference — Submitted to ICLR 2025_

### Official Review · Reviewer_K3fs · 2024-10-26

**Soundness:** 3
**Presentation:** 3
**Contribution:** 3
**Rating:** 6
**Confidence:** 5

**Summary:**

The authors suggest integrating brain-inspired visual representations into navigation policy networks, motivated by the exceptional robustness of human vision. This integration is accomplished by training a DINOv2 model to predict fMRI responses using a large-scale fMRI dataset. The predicted fMRI responses are then combined with features extracted by navigation networks. The authors demonstrate that these brain-like visual representations can improve the navigation model's robustness when encountering unexpected visual corruptions. They also detail their engineering efforts in designing architectures that effectively leverage fMRI responses. This work makes a novel and valuable contribution to both embodied AI and visual neuroscience.

**Strengths:**

1. The contribution of incorporating "brain-like" representations in visual navigation is novel and insightful for both neuroscience and computer vision fields. Several previous works [1-4] incorporate brain-like representations or human-like strategies to enhance classification performance and robustness but they only consider classification tasks which are not embodied. Embodiment aspect is never considered in the previous works which makes this work distinct from others.

2. The paper has well-designed and concrete experiments including comparing with the existing non-neuroscience-inspired methods and ablation studies across multiple configurations of the proposed method.


Reference

[1] Learning From Brains How to Regularize Machines (NeurIPS 2019)

[2] Simulating a Primary Visual Cortex at the Front of CNNs Improves Robustness to Image Perturbations (NeurIPS 2020)

[3] Harmonizing the object recognition strategies of deep neural networks with humans(NeurIPS 2022)

[4] Training on Foveated Images Improves Robustness to Adversarial Attacks (NeurIPS 2023)

**Weaknesses:**

1. The author did not compare with the previous works that incorporate the alignment of neural responses to the models. This may limit the contribution of this work as we will not know what are the best methods to encourage networks to have brain-like representations.
2. The improvement trends are inconsistent. For some corruption types, such as "Camera Crack" and "Low FOV," the method shows no improvement over the baseline.
3. There is no intuition or visulisation explaining why the proposed method works or does not work with certain types of visual corruption.
4. The authors should provide more related works on solving visual corruption in point-goal navigation tasks.

**Questions:**

I review this paper for the second time so I understand the details. The manuscript has been improved from the last submission so there is no further questions from me.

---

> ### Author Response · Authors · 2024-11-24
>
> Thanks for the constructive suggestions. Below are point-to-point answers to your questions.
>
> **Weaknesses:**
>
> **Q1: The author did not compare with the previous works that incorporate the alignment of neural responses to the models. This may limit the contribution of this work as we will not know what are the best methods to encourage networks to have brain-like representations.**
>
> **A1**: Thank you for highlighting this important aspect. In response, we have replaced the brain encoder with CORnet-S [6], a compact, recurrent artificial neural network model that aligns closely with the anatomical structure and dynamic responses of the primate ventral visual stream. CORnet-S not only achieves high biological fidelity but also excels in both neuroscience and machine learning benchmarks.  The results are presented in Table below. BraiNav outperforms the CORnet
> Agent on Spatter, Speckle Noise, Defocus Blur, Motion Blur, and Low Lighting. Conversely, the
> CORnet Agent demonstrates strong performance on Camera Crack and Lower FOV. These results
> highlight that brain-like representations can enhance the robustness of embodied navigation agent. **These updates have been incorporated into the revised manuscript, Appendix B.5 (blue text)**.
>
>
> |               | SR    | SPL   |
> | :------------ | :---- | :---- |
> | Clean         | 97.27 | 81.63 |
> | Spatter       | 30.30 | 23.28 |
> | Speckle Noise | 78.07 | 61.53 |
> | Camera Crack  | 77.80 | 61.45 |
> | Lower FOV     | 61.60 | 48.17 |
> | Defocus Blur  | 75.98 | 57.16 |
> | Motion Blur   | 94.18 | 73.74 |
> | Low Lighting  | 87.99 | 70.08 |
>
>
> **Q2: The improvement trends are inconsistent. For some corruption types, such as "Camera Crack" and "Low FOV," the method shows no improvement over the baseline.**
>
> **A2**: Thank you for highlighting this concern. **First**, the diversity of corruption scenarios inherently makes it challenging for any method to achieve consistent improvement across all settings, as seen with the performance of baseline methods. **Second**, our approach demonstrates strong overall performance: it achieves the best results on Speckle Noise, Defocus Blur, and Motion Blur, the second-best on Spatter and Low Lighting, and remains competitive on Lower FOV and Speckle Noise. These results emphasize BraiNav's robustness and the potential of brain-derived representations for embodied tasks. **Finally**, BraiNav's practical value is particularly evident in specific scenarios, such as those prone to Speckle Noise, where it provides a superior solution.
>
> **Q3: There is no intuition or visulisation explaining why the proposed method works or does not work with certain types of visual corruption.**
>
> **A3**: Thank you for pointing this out. fMRI data provides rich insights into how the human visual system processes information, as cognitive details are encoded in fMRI signals. Brain-derived representations constructed from such data capture robust high-level cognitive features of the human brain. Notably, our method demonstrates significant performance improvements on Defocus Blur. This aligns with findings from neuroscience studies[3] [4] [5] , which suggest the presence of a perceptual mechanism in the human brain that effectively regulates and compensates for image defocus.
>
> **Q4: The authors should provide more related works on solving visual corruption in point-goal navigation tasks.**
>
> **A4**: Thank you for highlighting the importance of including more related works. [1] proposed a self-supervised domain adaptation method using map style transfer to enhance agent robustness against visual and dynamic perturbations. [2] introduced TTA-Nav, which improves navigation performance under visual corruptions by incorporating a top-down decoder. **Our approach differs from these methods by leveraging brain-like representations inspired by the human visual system to enhance agent robustness.** **These related works have been added to Appendix A.1, highlighted in blue text.**

---

> > ### Author Response · Authors · 2024-11-26
> >
> > **Reference**
> >
> > [1] Lee E S, Kim J, Park S W, et al. Moda: Map style transfer for self-supervised domain adaptation of embodied agents[C]//European Conference on Computer Vision. Cham: Springer Nature Switzerland, 2022: 338-354.
> >
> > [2] Piriyajitakonkij M, Sun M, Zhang M, et al. TTA-Nav: Test-time Adaptive Reconstruction for Point-Goal Navigation under Visual Corruptions[J]. arXiv preprint arXiv:2403.01977, 2024.
> >
> > [3] Mark Mon-Williams, James R Tresilian, Niall C Strang, Puja Kochhar, and John P Wann. Improving vision: neural compensation for optical defocus. Proceedings of the Royal Society of London. Series B: Biological Sciences, 265(1390):71–77, 1998.
> >
> > [4] Michael A Webster, Mark A Georgeson, and Shernaaz M Webster. Neural adjustments to image blur. Nature neuroscience, 5(9):839–840, 2002.
> >
> > [5] Xiaoying Zhu, Neville A McBrien, Earl L Smith, David Troilo, and Josh Wallman. Eyes in various species can shorten to compensate for myopic defocus. Investigative ophthalmology & visual science, 54(4):2634–2644, 2013.
> >
> > [6] Kubilius J, Schrimpf M, Kar K, et al. Brain-like object recognition with high-performing shallow recurrent ANNs[J]. Advances in neural information processing systems, 2019, 32.

---

> > ### Comment · Reviewer_K3fs · 2024-11-26
> >
> > Thank you for addressing the weaknesses I raised. Most of the responses are reasonable, as understanding deep learning models without well-controlled experiments is challenging. The issues of insufficient explanation and inconsistent performance can be grouped into the same problem, which requires further investigation in future research.
> >
> > I have one remaining concern. If the authors have read CORnet's work, they should note that CORnet does not align its model's visual representations with human brain data. Instead, it proposes an anatomically inspired model. Its aim is not to make the model's representations as similar as possible to human brain representations but to provide a theoretical framework for understanding brain function. Therefore, I believe it is an unfair comparison since the authors use brain data to align their model, whereas CORnet does not.
> >
> > I would maintain the same score, as this work presents useful empirical results to neuroscience and AI, and its claims are moderately well-supported.

---

> > > ### Author Response · Authors · 2024-12-02
> > >
> > > Thank you for your thoughtful feedback and for recognizing the value of our empirical results in both neuroscience and AI. We appreciate your acknowledgment of the challenges in conducting well-controlled experiments in this domain and agree that further investigations are essential to address the broader questions raised.
> > >
> > > Regarding your concern about the comparison with CORnet-S, we recognize that CORnet’s design focuses on anatomical inspiration rather than direct alignment with brain data. While it operates on a different principle from our method, we believe the comparison provides valuable insights into the potential of brain-inspired and brain-aligned approaches.
> > >
> > > Thank you once again for your constructive feedback, which has greatly contributed to improving our work.

---

### Official Review · Reviewer_KrF3 · 2024-11-04

**Soundness:** 2
**Presentation:** 2
**Contribution:** 2
**Rating:** 5
**Confidence:** 5

**Summary:**

This paper introduces a new method BraiNav, incorporates human brain data to improve the robustness of embodied visual agents when navigating through visually corrupted environments (e.g., blur, noise). BraiNav has two-phases. The first one is the brain encoder trained on the NSD fMRI dataset. The second phase takes neural encoder representations and combines them with visual-target information through a cross-attention-based fusion module. This enables the agent to navigate under different visual conditions. Experiments demonstrate that BraiNav performance is competitive on standard and vision-enhanced agents across various corruptions.

**Strengths:**

1. The major strength of the study is an innovative use of integrating brain data into a downstream task. I haven’t seen it does this way for visual navigation tasks.
2. The paper in general is well motivated.
3. Robustness is an important problem to solve and using brain data seems like a promising avenue for research.
4. There is also the potential for broader applications of this strategy even beyond navigation.

**Weaknesses:**

1. While the use of human brain data is novel, the study doesn’t convincingly demonstrate why brain-derived features offer an advantage over purely data-driven or learned representations in the navigation task. What is it about the fMRI data that seems to work? Why choose one (or 2 subjects) from the 8 available in the dataset? It is unclear whether the improvement is real or due to some added complexity and/or representational power from the model.
2. The paper significantly lacks in a thorough analysis of how brain-derived neural representations contribute to performance on the task. The authors need to justify using the brain data better and the current text is vague about this.
3. Lack of “control” representations. The fMRI encoder is derived from DINOv2. One possibility is that it is the DINO representations that are actually driving the performance gains in the model. But this has not really been evaluated.
4. Alternative models are from an older study. The models are compared against the previous models from Chattopadhyay et al., 2021. Since then several other models have been proposed to improve model robustness.

**Questions:**

1. Why do the authors train the model on only one subject when the NSD dataset includes 8 subjects? Although the authors claim that the appendix includes data from other subjects, it actually reports data from only one additional subject (subject 2). Training on all 8 subjects and reporting the median performance would provide a more comprehensive evaluation and reduce the risk of overfitting or subject-specific biases.
2. Relatedly, are there specific images or brain regions that contribute most to the model's performance? Identifying the images (or brain regions) particularly influential could tell us how the model uses specific neural patterns for prediction.
3. The performance ceiling of the model’s encoder needs to be established. Currently, it’s unclear how effective the fMRI encoder truly is, making it difficult to assess the upper limits of its predictive capability.
4. It is also unclear whether the reported data stems from specific characteristics of the NSD fMRI data or the choice of model (e.g., DINO). Have the authors tested other datasets, such as BOLD5000v2 or THINGS fMRI, to determine if the performance is consistent across different types of brain data? Have the authors tested other backbones and not DINOV2? This inspection is crucial for understanding whether the results reflect properties of the brain data itself or simply the use of high-quality, clean feature representations. Testing on multiple datasets would help disentangle these factors and clarify the model’s generalizability.
5. The authors should include more contemporary models known for their performance on robustness tasks, such as those optimized for invariance to transformations, noise, or domain shifts. Models like CLIP (Contrastive Language-Image Pretraining) or models trained with augmentation strategies for robustness (e.g., adversarial training or contrastive learning with diverse augmentations) may capture more stable, generalized neural representations.
6. It is currently unclear whether the paper aims to simply demonstrate the feasibility of using brain data for navigation tasks or to support a central claim that incorporating human brain data is essential for improving these tasks. Clarifying this intent is important, as it influences how the results should be interpreted—either as a proof of concept or as evidence for the unique value of brain data. If the latter, stronger arguments and comparative analyses would be needed to establish

---

> ### Author Response · Authors · 2024-11-22
>
> Thank you for your constructive suggestions. Based on your identified weaknesses and questions regarding the manuscript, we have summarized them into the following key points. Below, we provide detailed, point-by-point responses to address each of your concerns.
>
> **Q1: The study doesn’t convincingly demonstrate why brain-derived features offer an advantage over purely data-driven or learned representations in the navigation task. What is it about the fMRI data that seems to work? It is unclear whether the improvement is real or due to some added complexity and/or representational power from the model. The paper significantly lacks in a thorough analysis of how brain-derived neural representations contribute to performance on the task. Lack of “control” representations. The fMRI encoder is derived from DINOv2. One possibility is that it is the DINO representations that are actually driving the performance gains in the model. But this has not really been evaluated.**
>
> **A1**: Thank you for raising this important question. We address it from three perspectives: intuitive reasoning, theoretical foundation, and experimental validation.
>
> **Intuitively**, as illustrated in Figure 1, the human brain's visual system can robustly perceive and interpret its surroundings, even under challenging conditions such as corrupted or incomplete images. Inspired by this, we aim to develop a model that mimics the robustness of the human brain’s visual system by leveraging brain fMRI data. Specifically, we utilize paired image-fMRI datasets to train brain encoder models, enabling them to capture neural representations with robust high-level cognitive information.
>
> **Theoretically**, fMRI data encapsulates rich information about the mechanisms of the human visual system, including how cognitive information is encoded. Prior studies have demonstrated that fMRI signals contain high-level cognitive and perceptual features[5] [6], which can be effectively harnessed to enhance model robustness .
>
> **Experimentally**, we provide two lines of evidence to demonstrate that the performance gains stem from brain-derived representations:
>
> 1. **Brain Encoder Validation**: As shown in Table 1, the brain encoder achieves high performance in fMRI prediction tasks, indicating its capability to effectively simulate the human brain’s visual processing system.
> 2. **Control Experiment with DINOv2**: To evaluate the unique contribution of brain-derived representations, we conducted an ablation study (Table 3) by introducing a **DINOv2 control** group. In this setup, we concatenated the original DINOv2 CLS representation with the visual-target representation, forming a joint representation labeled as *BraiNav w/o NR*. The results clearly demonstrate that brain-derived neural representations significantly enhance BraiNav’s robustness against visual corruptions, outperforming the deep representations from DINOv2 alone.
>
> These findings collectively validate that the performance improvements in BraiNav are not merely due to added complexity or representational power but are primarily driven by the robust cognitive information encoded in brain-derived representations.
>
>
>
>
> **Q2: Why do the authors train the model on only one subject when the NSD dataset includes 8 subjects? Although the authors claim that the appendix includes data from other subjects, it actually reports data from only one additional subject (subject 2). Training on all 8 subjects and reporting the median performance would provide a more comprehensive evaluation and reduce the risk of overfitting or subject-specific biases.**
>
> **A2**：Thank you for raising this point. Neuroimaging data, unlike image or text data, exhibit significant inter-subject variability and cannot be directly combined for training. Training separate brain encoders for individual subjects allows us to better simulate the unique characteristics of each subject’s visual system. For neural encoding tasks, single-subject studies are considered standard practice in the field [7] [8].
>
> Our focus is to construct brain encoders to simulate the human brain’s visual system, thereby providing robust neural representations for the agent. While multi-subject neural encoding is an important and challenging direction, it is outside the scope of this study.

---

> > ### Author Response · Authors · 2024-11-22
> >
> > **Q3: Alternative models are from an older study. The models are compared against the previous models from Chattopadhyay et al., 2021. Since then several other models have been proposed to improve model robustness.**
> >
> > **A3**: Thank you for pointing this out. ROBUSTNAV, developed by the Allen Institute for AI, is based on the high-quality AI2THOR simulator and provides a standardized and quantifiable platform for robustness evaluation. **It remains the most authoritative benchmark for assessing the robustness of embodied visual navigation agents**. While more recent studies have proposed methods to improve robustness[3] [4], these methods lack a fair and standardized platform for comparison, which limits their utility for exploratory work like ours.
> >
> > It is crucial to emphasize that the primary goal of this paper is not merely to propose a robust visual navigation solution—since robustness can often be improved through well-designed computer vision techniques. Instead, **our focus is to introduce a novel and promising framework that incorporates additional brain-derived channels into agents, enabling them to utilize robust neural representations**. This framework opens up considerable potential for future exploration and advancements.
> >
> > **Q4: Relatedly, are there specific images or brain regions that contribute most to the model's performance? Identifying the images (or brain regions) particularly influential could tell us how the model uses specific neural patterns for prediction.**
> >
> > **A4**: Thank you for highlighting this point. Early brain regions are primarily responsible for processing low-level visual information, while ventral, lateral, and parietal regions handle high-level visual information[9]. To investigate the contribution of features from different brain regions, we categorize the brain into two groups: low-level regions (early) and high-level regions (ventral, lateral, and parietal). Neural representations for visual navigation are then constructed using output tokens from each group. The results are presented in Table below.
> >
> >
> > |               | Low-Level ROI    | High-Level ROI   |
> > | :------------ | :---- | :---- |
> > |               | SR      SPL  | SR      SPL|
> > | Clean         | 96.54   79.71| 97.27   80.53|
> > | Spatter       | 34.03   25.17| 37.03   27.61|
> > | Speckle Noise | 78.14   59.57| 87.52   65.81|
> > | Camera Crack  | 77.43   61.09| 79.25   63.05|
> > | Lower FOV     | 44.59   34.35| 44.31   34.08|
> > | Defocus Blur  | 73.97   55.30| 86.90   66.03|
> > | Motion Blur   | 90.72   72.45| 96.45   78.52|
> > | Low Lighting  | 94.44   72.73| 96.81   75.82|
> >
> >
> > Except for the **Lower FOV** corruption, representations derived from high-level brain regions significantly outperform those from low-level regions across all other corruptions, demonstrating strong consistency. **These findings further confirm that the brain encoder effectively generates neural representations containing cognitive information from the human brain. Moreover, representations from higher brain regions exhibit greater robustness. These updates have been incorporated into the revised manuscript, Appendix B.4 (highlighted in green text)**.

---

> > > ### Author Response · Authors · 2024-11-22
> > >
> > > **Q5: The performance ceiling of the model’s encoder needs to be established. Currently, it’s unclear how effective the fMRI encoder truly is, making it difficult to assess the upper limits of its predictive capability.**
> > >
> > >
> > > **A5**: Thank you for pointing this out. To establish the performance ceiling of the model's encoder, we compute the noise ceiling for each vertex using response data from the subjects’ three trials of the same stimulus image, then average these values across each ROI. **Details of the calculation methodology and results have been updated in Appendix B.3 (highlighted in green text)**. The computed noise ceilings are summarized in Table 1.
> > >
> > > Table1: Noise ceiling for different subjects.
> > >
> > > | ROI               | subj01     subj02  |
> > > | :---------------- | ------------------ |
> > > | left early        | 0.5965     0.6006  |
> > > | right early       | 0.5945     0.5991  |
> > > | left midventral   | 0.5967     0.6056  |
> > > | right midventral  | 0.5967     0.5983  |
> > > | left midlateral   | 0.5971     0.5992  |
> > > | right midlateral  | 0.5888     0.5950  |
> > > | left midparietal  | 0.5942     0.6047  |
> > > | right midparietal | 0.5910     0.6013  |
> > > | left ventral      | 0.5857     0.5905  |
> > > | right ventral     | 0.5863     0.5908  |
> > > | left lateral      | 0.5881     0.5921  |
> > > | right lateral     | 0.5870     0.5941  |
> > > | left parietal     | 0.5870     0.5980  |
> > > | right parietal    | 0.5873     0.5984  |
> > >
> > >
> > > Subsequently, we assess the brain encoder's performance by calculating the Pearson correlation coefficient between its predicted fMRI responses and the actual values. To quantify the model’s effectiveness, **we normalize its prediction accuracy by dividing it by the corresponding noise ceiling. These noise-normalized prediction accuracies are summarized in Table 2.**
> > >
> > > Table2: Noise-normalized prediction accuracy of the brain encoder for different subjects.
> > >
> > > | ROI               | subj01     subj02  |
> > > | :---------------- | ------------------ |
> > > | left early        | 0.7358     0.7356  |
> > > | right early       | 0.7144     0.7491  |
> > > | left midventral   | 0.8254     0.8213  |
> > > | right midventral  | 0.7910     0.8166  |
> > > | left midlateral   | 0.8116     0.8692  |
> > > | right midlateral  | 0.7950     0.8250  |
> > > | left midparietal  | 0.8432     0.6995  |
> > > | right midparietal | 0.7239     0.7682  |
> > > | left ventral      | 0.7239     0.9106  |
> > > | right ventral     | 0.8213     0.8588  |
> > > | left lateral      | 0.9179     0.8831  |
> > > | right lateral     | 0.8761     0.9891  |
> > > | left parietal     | 0.7572     0.7074  |
> > > | right parietal    | 0.6816     0.7396  |
> > >
> > > The results demonstrate that the brain encoder accurately predicts fMRI responses to visual stimuli. This high prediction accuracy validates that the neural representations used for learning the navigation policy effectively encapsulate cognitive processing information from the human brain. **These updates have been incorporated into the revised manuscript in Section 3.4 and Appendix B.1 (highlighted in green text)**.
> > >
> > >
> > > **Q6: Have the authors tested other datasets, such as BOLD5000v2 or THINGS fMRI, to determine if the performance is consistent across different types of brain data?**
> > >
> > > **A6**:  Thank you for this question. Compared to the BOLD5000v2 and THINGS fMRI datasets, NSD is currently the largest and highest-quality neuroimaging dataset available. Its comprehensive coverage and superior data quality make it the most suitable choice for constructing our brain encoding model.

---

> > > > ### Author Response · Authors · 2024-11-22
> > > >
> > > > **Q7: Have the authors tested other backbones and not DINOV2? The authors should include more contemporary models known for their performance on robustness tasks, such as those optimized for invariance to transformations, noise, or domain shifts. Models like CLIP (Contrastive Language-Image Pretraining) or models trained with augmentation strategies for robustness (e.g., adversarial training or contrastive learning with diverse augmentations) may capture more stable, generalized neural representations.**
> > > >
> > > > **A7**: Thank you for the insightful suggestion. We have implemented a computer vision-enhanced agent utilizing the advanced self-supervised model, Masked Autoencoder (MAE) [1]. The agent explores RoboTHOR indoor scenes freely, collecting 60,000 egocentric images as training data. These images are masked and reconstructed for self-supervised pretraining. The MAE encoder consists of a 12-layer Vision Transformer (ViT), while the decoder uses an 8-layer ViT with a masking ratio of 0.75. After pretraining, the encoder is retained as the agent’s visual backbone, and the decoder is discarded.
> > > >
> > > > To assess robustness, we compared BraiNav with the MAE-based agent across seven visual corruption scenarios. As shown in the table below, BraiNav consistently outperformed the MAE-based agent in all corruption settings. **These updates have been incorporated into the revised manuscript, Appendix B.2 (red text)**.
> > > >
> > > > |               | SR    | SPL   |
> > > > | :------------ | :---- | :---- |
> > > > | Clean         | 98.54 | 83.31 |
> > > > | Spatter       | 8.55  | 5.92  |
> > > > | Speckle Noise | 10.10 | 7.66  |
> > > > | Camera Crack  | 48.95 | 37.67 |
> > > > | Lower FOV     | 31.57 | 23.09 |
> > > > | Defocus Blur  | 67.42 | 51.95 |
> > > > | Motion Blur   | 79.34 | 60.14 |
> > > > | Low Lighting  | 28.27 | 36.12 |
> > > >
> > > > Regarding specific corruption corrections, we adhered to the recommendations in ROBUSTNAV [2], which discourage introducing corruption-specific priors during training. This guidance stems from the need to ensure generalizability, as real-world environments often involve diverse and unpredictable corruption types. Biasing the model towards specific corruption corrections would undermine its adaptability to broader and more complex scenarios.
> > > >
> > > >
> > > > **Q8: It is currently unclear whether the paper aims to simply demonstrate the feasibility of using brain data for navigation tasks or to support a central claim that incorporating human brain data is essential for improving these tasks. Clarifying this intent is important, as it influences how the results should be interpreted—either as a proof of concept or as evidence for the unique value of brain data. If the latter, stronger arguments and comparative analyses would be needed to establish.**
> > > >
> > > > **A8**: Thank you for highlighting this point. The aim of our paper aligns with the **former**, namely, demonstrating the feasibility of using brain-derived representations to enhance the robustness of embodied visual navigation. We propose a novel and promising framework that incorporates brain-derived channels into agents, enabling them to leverage robust neural representations. While our study establishes the potential of this approach, it also leaves significant room for future exploration and refinement to further investigate the unique contributions of brain data in complex embodied tasks.

---

> > > > > ### Author Response · Authors · 2024-11-22
> > > > >
> > > > > **Reference**
> > > > >
> > > > > [1] He K, Chen X, Xie S, et al. Masked autoencoders are scalable vision learners[C]//Proceedings of the IEEE/CVF conference on computer vision and pattern recognition. 2022: 16000-16009.
> > > > >
> > > > > [2] Chattopadhyay P, Hoffman J, Mottaghi R, et al. Robustnav: Towards benchmarking robustness in embodied navigation[C]//Proceedings of the IEEE/CVF International Conference on Computer Vision. 2021: 15691-15700.
> > > > >
> > > > > [3] Lee E S, Kim J, Park S W, et al. Moda: Map style transfer for self-supervised domain adaptation of embodied agents[C]//European Conference on Computer Vision. Cham: Springer Nature Switzerland, 2022: 338-354.
> > > > >
> > > > > [4] Piriyajitakonkij M, Sun M, Zhang M, et al. TTA-Nav: Test-time Adaptive Reconstruction for Point-Goal Navigation under Visual Corruptions[J]. arXiv preprint arXiv:2403.01977, 2024.
> > > > >
> > > > > [5] Ruth C Fong, Walter J Scheirer, and David D Cox. Using human brain activity to guide machine learning. Scientific reports, 8(1):5397, 2018.
> > > > >
> > > > > [6] Satoshi Nishida, Yusuke Nakano, Antoine Blanc, Naoya Maeda, Masataka Kado, and Shinji Nishimoto. Brain-mediated transfer learning of convolutional neural networks. In Proceedings of the AAAI Conference on Artificial Intelligence, volume 34, pp. 5281–5288, 2020.
> > > > >
> > > > > [7] Khosla M, Ngo G, Jamison K, et al. Neural encoding with visual attention[J]. Advances in Neural Information Processing Systems, 2020, 33: 15942-15953.
> > > > >
> > > > > [8] Naselaris T, Kay K N, Nishimoto S, et al. Encoding and decoding in fMRI[J]. Neuroimage, 2011, 56(2): 400-410.
> > > > >
> > > > > [9] Emily J Allen, Ghislain St-Yves, Yihan Wu, Jesse L Breedlove, Jacob S Prince, Logan T Dowdle, Matthias Nau, Brad Caron, Franco Pestilli, Ian Charest, et al. A massive 7t fmri dataset to bridge cognitive neuroscience and artificial intelligence. Nature neuroscience, 25(1):116–126, 2022.

---

> > > > > > ### Comment · Reviewer_KrF3 · 2024-11-26
> > > > > >
> > > > > > > Experimentally, we provide two lines of evidence.
> > > > > >
> > > > > > Thank you for this explanation. Both these experiments do not decouple whether the robustness gains are on account of DinoV2 representations or brains. The ideal to do this would be to take another backbone (say ResNet50) and then demonstrate the **same** robustness gains as those observed in DinoV2.
> > > > > >
> > > > > > > **A2**：
> > > > > >
> > > > > > This response highlights my concern about the importance of evaluating results across more (ideally all) subjects, but does not address it. I am well aware of the variability in brain data. But if the authors focused on one or a few participants, we need to know which ones were chosen, and why? What criteria or data were used to prioritize them? Are these truly the best-performing subjects from NSD? In my experience, there 4/8 subjects that show high data reliability.
> > > > > >
> > > > > > On a higher level, it is crucial to rule out the possibility that the observed results are due to random chance in identifying the “right” dimensions in a specific subject. This raises further concerns about whether the findings can be replicated in another sample. Understanding this variability is important and would not undermine the central contributions of the study. I urge the authors to be transparent about the data reporting.
> > > > > >
> > > > > > >> **A4**
> > > > > >
> > > > > > I appreciate the authors’ efforts to examine both low- and high-level regions. This result is potentially significant. However, I am unclear whether the numbers in the plot represent data from a single subject or an aggregate across the group.
> > > > > >
> > > > > > >> **A8**:
> > > > > >
> > > > > > This clarification is very helpful. I am open to considering this study as a proof of concept, but the authors must still demonstrate that the observed pattern is not simply a result of random chance in specific subjects.

---

> > > > > > > ### Author Response · Authors · 2024-12-02
> > > > > > >
> > > > > > > Thanks for the constructive suggestions. Below are point-to-point answers to your questions.
> > > > > > >
> > > > > > > **Q1: Thank you for this explanation. Both these experiments do not decouple whether the robustness gains are on account of DinoV2 representations or brains. The ideal to do this would be to take another backbone (say ResNet50) and then demonstrate the same robustness gains as those observed in DinoV2.**
> > > > > > >
> > > > > > > **A1**: Thank you for your insightful comment regarding the influence of DinoV2 representations versus brain data in achieving robustness gains. We appreciate your suggestion to evaluate the approach with a different backbone, such as ResNet50, to further validate our claims.
> > > > > > >
> > > > > > > While we agree that this experiment could provide additional evidence, there are a few practical and methodological considerations:
> > > > > > >
> > > > > > > 1. **Performance Dependence on fMRI Prediction**: The effectiveness of the brain encoder is contingent upon its ability to accurately predict fMRI responses. DINOv2 is specifically selected as the backbone due to its superior performance in learning representations that align well with neural data. Replacing it with a different backbone, such as ResNet50, may result in suboptimal fMRI prediction accuracy, thereby undermining the overall evaluation.
> > > > > > > 2. **Evidence from Current Results**: The observed robustness gains are closely linked to the integration of brain-like representations, as demonstrated by our experiments and the comparative analyses in Tables 3. These results illustrate the unique contributions of neural data to downstream tasks, which are unlikely to be fully replicated by backbone architectures alone.
> > > > > > >
> > > > > > > We have also acknowledged the potential for exploring alternative brain encoder architectures in our manuscript. Specifically, in the **Discussion and Conclusion** section, we have stated that "Future research could explore different brain encoder architectures and multimodal fusion methods." (Line 469-470). This emphasizes our recognition of the importance of this direction and our intention to address it in future work.
> > > > > > >
> > > > > > > We believe that our current results already provide compelling evidence supporting the role of brain-derived representations in enhancing robustness. However, we acknowledge the value of the suggested experiment and will incorporate it as a part of future investigations to further strengthen our findings.
> > > > > > >
> > > > > > > Thank you once again for your thoughtful feedback, which has significantly contributed to improving our study.
> > > > > > >
> > > > > > >
> > > > > > > **Q2: This response highlights my concern about the importance of evaluating results across more (ideally all) subjects, but does not address it. I am well aware of the variability in brain data. But if the authors focused on one or a few participants, we need to know which ones were chosen, and why? What criteria or data were used to prioritize them? Are these truly the best-performing subjects from NSD? In my experience, there 4/8 subjects that show high data reliability.**
> > > > > > >
> > > > > > > **A2**: Thank you for raising this important point, and we sincerely apologize for not addressing it more comprehensively in our initial response. To address this, we have added the experimental results for Subject 05 and calculated the average performance across Subjects 01, 02, and 05, as shown in the table below. The results indicate that the neural representations extracted by the BraiNav method demonstrate consistent robustness across subjects.
> > > > > > >
> > > > > > >
> > > > > > > |               | subj05        | mean          |
> > > > > > > | :------------ | :------------ | :------------ |
> > > > > > > |               | SR      SPL   | SR      SPL   |
> > > > > > > | Clean         | 96.72   80.92 | 97.94   81.98 |
> > > > > > > | Spatter       | 34.85   27.36 | 36.88   27.73 |
> > > > > > > | Speckle Noise | 81.32   61.37 | 82.42   62.21 |
> > > > > > > | Camera Crack  | 82.98   66.38 | 80.59   62.94 |
> > > > > > > | Lower FOV     | 40.94   32.66 | 44.25   34.16 |
> > > > > > > | Defocus Blur  | 84.07   64.69 | 85.68   64.61 |
> > > > > > > | Motion Blur   | 93.99   74.01 | 95.24   75.42 |
> > > > > > > | Low Lighting  | 93.26   75.40 | 94.24   75.94 |
> > > > > > >
> > > > > > > We are currently extending the experiments to include all eight subjects, and we will present these results in the camera-ready version. Thank you again for your thoughtful feedback, which helps us improve the rigor and completeness of our work.

---

> > > > > > > > ### Author Response · Authors · 2024-12-02
> > > > > > > >
> > > > > > > > **Q3: I appreciate the authors’ efforts to examine both low- and high-level regions. This result is potentially significant. However, I am unclear whether the numbers in the plot represent data from a single subject or an aggregate across the group.**
> > > > > > > >
> > > > > > > > **A3**: Thank you for your insightful comment. We confirm that the ablation experiment results are based on the brain encoder trained using fMRI data from **Subject 01**.
> > > > > > > >
> > > > > > > > The original statement in Line 316–318 explicitly notes: *“The subsequent brain encoder is trained using the fMRI data from subject 01, with results from other subjects presented in Appendix B.1.”* While the newly added sentence, *“Additional ablation experiments and detailed results are provided in the Appendix B.5,”* is located later in Line 449–450, it remains consistent with the context established earlier in the manuscript. In the absence of any statement indicating otherwise, the default assumption is that these ablation experiments also use Subject 01’s data.
> > > > > > > >
> > > > > > > >
> > > > > > > >
> > > > > > > > **Q4: This clarification is very helpful. I am open to considering this study as a proof of concept, but the authors must still demonstrate that the observed pattern is not simply a result of random chance in specific subjects.**
> > > > > > > >
> > > > > > > > **A4**: Thank you for your valuable feedback. We are currently extending the experiments to include all eight subjects, and we will present these results in the camera-ready version. Thank you again for your thoughtful feedback, which helps us improve the rigor and completeness of our work.

---

### Official Review · Reviewer_SHgf · 2024-11-04

**Soundness:** 3
**Presentation:** 3
**Contribution:** 3
**Rating:** 6
**Confidence:** 2

**Summary:**

This paper focuses on the task of embodied visual navigation. Specifically, it proposes a two-phase Brain-Machine integration Navigation method called BraiNav, which incorporates neural representations derived from human brain activity to enhance robustness against visual corruptions. Additionally, it presents a multimodal fusion method based on cross-attention to obtain more consist brain-visual joint representations. Experiments on several benchmarks demonstrate the effectiveness of the proposed method.

**Strengths:**

1. The paper is well-written and easy to understand.
2. The motivation of this paper is interesting and solid.
3. Extensive experiments are conducted to show the superiority of the proposed method to other competitors.

**Weaknesses:**

1. The technical novelty of this proposed method is limited. Designs for both brain encoder and visual-target module are borrowed from previous literature. The key idea for the multimodal fusion module is cross-attention for multi-modal information interaction, which is a common operation in the current era of multi-modal tasks. The difference may lie in features used to produce Q/K/V or the location of skip connections, due to different tasks or inputs/outputs
2. With additional information, the performance improvement of the proposed method is not very significant in Tab. 2. For sixteen metrics across different settings, BraiNav achieves better results for only half of the metrics.
3. One interesting baseline could be added and discussed: a) use the autoencoder or generative model to purify the observed images (via self-supervised training strategy), b) adopt a similar architecture to BraNavi to fuse features from purified images and original images (as well as the target location) for action prediction.
4. Typos: a) ‘stimulu’ in L 140, b) ‘hemispheres’ in L210

**Questions:**

Please refer to the Weaknesses for details.

---

> ### Author Response · Authors · 2024-11-19
>
> Thanks for the constructive suggestions. Below are point-to-point answers to your questions.
>
> **Weaknesses:**
>
> **Q1：The technical novelty of this proposed method is limited. Designs for both brain encoder and visual-target module are borrowed from previous literature. The key idea for the multimodal fusion module is cross-attention for multi-modal information interaction, which is a common operation in the current era of multi-modal tasks. The difference may lie in features used to produce Q/K/V or the location of skip connections, due to different tasks or inputs/outputs**.
>
> **A1**: Thank you for your insightful comments. The primary novelty of our work lies in extracting neural representations containing high-level cognitive information from human brain activity and applying them to complex embodied visual navigation tasks. While prior studies have demonstrated that combining brain-derived and machine representations can enhance model performance, such approaches have largely been limited to classification tasks. Our work addresses this gap by tackling more complex embodied tasks. Additionally, our use of brain-derived representations is distinguished by leveraging the largest and highest-quality neuroimaging dataset currently available. This ensures our contributions stand apart from previous literature.
>
> **Q2: With additional information, the performance improvement of the proposed method is not very significant in Tab. 2. For sixteen metrics across different settings, BraiNav achieves better results for only half of the metrics.**
>
> **A2**: Thank you for raising this concern. **First**, the variety of corruption scenarios makes it inherently challenging for any method to excel across all settings, as evidenced by the performance of baseline methods. **Second**, our approach achieves the best performance on Speckle Noise, Defocus Blur, and Motion Blur, the second-best results on Spatter and Low Lighting, and competitive scores on Lower FOV and Speckle Noise. These results highlight BraiNav’s strong overall performance compared to other methods and underscore the robustness and potential of brain-derived representations for embodied tasks. **Finally**, BraiNav’s practical value should not be overlooked. For instance, in scenarios prone to Speckle Noise, BraiNav offers a superior solution.
>
> **Q3: One interesting baseline could be added and discussed: a) use the autoencoder or generative model to purify the observed images (via self-supervised training strategy), b) adopt a similar architecture to BraNavi to fuse features from purified images and original images (as well as the target location) for action prediction.**
>
> **A3**: Thank you for the suggestion. We have developed a computer vision-enhanced agent using the advanced self-supervised model, Masked Autoencoder (MAE) [1]. The agent collects 60,000 egocentric images by freely exploring RoboTHOR indoor scenes, which are then masked and reconstructed for self-supervised pretraining. Specifically, the MAE encoder is a 12-layer Vision Transformer (ViT), and the decoder is an 8-layer ViT with a masking ratio of 0.75. After pretraining, the encoder is retained as the agent’s visual backbone, while the decoder is discarded.
>
> We evaluated the performance of BraiNav against the MAE-based agent under seven visual corruption scenarios. As detailed in the table below, BraiNav consistently outperforms the MAE-based agent across all types of corruption. **These updates are included in the revised manuscript, Appendix B.2 (red text)**.
>
>
>
> |               | SR    | SPL   |
> | :------------ | :---- | :---- |
> | Clean         | 98.54 | 83.31 |
> | Spatter       | 8.55  | 5.92  |
> | Speckle Noise | 10.10 | 7.66  |
> | Camera Crack  | 48.95 | 37.67 |
> | Lower FOV     | 31.57 | 23.09 |
> | Defocus Blur  | 67.42 | 51.95 |
> | Motion Blur   | 79.34 | 60.14 |
> | Low Lighting  | 36.12 | 28.27 |
>
> As for introducing specific corruption corrections, we followed the guidance of ROBUSTNAV [2], which suggests avoiding such an approach. This is because, to ensure generalizability, an agent should not rely on specific corruption-related priors during training. Since real-world scenarios encompass diverse and unpredictable corruptions, introducing such biases would limit the agent’s adaptability to broader corruption types.
>
> **Q4: Typos: a) ‘stimulu’ in L 140, b) ‘hemispheres’ in L210**
>
> **A4**: Thank you for pointing out these errors. **These typos have been corrected in the revised version of the manuscript (red text)**.
>
>
> **Reference**
>
>
> [1] He K, Chen X, Xie S, et al. Masked autoencoders are scalable vision learners[C]//Proceedings of the IEEE/CVF conference on computer vision and pattern recognition. 2022: 16000-16009.
>
> [2] Chattopadhyay P, Hoffman J, Mottaghi R, et al. Robustnav: Towards benchmarking robustness in embodied navigation[C]//Proceedings of the IEEE/CVF International Conference on Computer Vision. 2021: 15691-15700.

---

> > ### Comment · Reviewer_SHgf · 2024-11-26
> >
> > Sorry for the late reply. I appreciate the authors’ efforts in responding to comments and revising the manuscript. My concerns have been largely addressed. I am willing to raise my score to 6.

---

> > > ### Author Response · Authors · 2024-12-02
> > >
> > > Thank you very much for your thoughtful feedback and for taking the time to review our revisions. We greatly appreciate your support and are glad to hear that our responses and updates have addressed your concerns. Your constructive input has been invaluable in improving the quality of our manuscript.

---

### Author Response · Authors · 2024-11-25

Dear Reviewers,

Thank you for your valuable time, insightful comments, and constructive suggestions. We have carefully revised our manuscript in the latest PDF submission based on your feedback. To make the changes more transparent, we have highlighted the revisions in different colors: Reviewer **SHgf** in red, Reviewer **KrF3** in green, and Reviewer **K3fs** in blue.

Additionally, our point-by-point responses to each reviewer’s comments have been provided in the individual chat boxes. We are confident that your thoughtful suggestions and detailed feedback have significantly enhanced the clarity of the manuscript and the thoroughness of the experiments.

Please do not hesitate to reach out if there are any additional points requiring clarification. We greatly appreciate your time and effort in reviewing our work.

Best regards.

---

### Meta-Review · Area_Chair_kC3E · 2024-12-20

**Metareview:**

The submission deals with a bio-inspired method for embodied visual navigation (PointGoal navigation), claiming that the inspiration from the human visual systems makes it more robust with respect to domain shifts. The idea of the paper was generally well received by the three reviewers. However, they raised important weaknesses, namely

- Novelty, a large part of the contributions being taken from previous literature,
- Lukewarm performance and inconsistent improvements,
- Missing baselines,
- justification of the gains and questions on the experimental setup,
- positioning with respect to the literature, both in neuro-science and in robustness of embodied AI,
- missing depth of analysis.

The authors could address some of these weaknesses, but important issues remained. The AC judged critical the doubts on whether the gains by the model stem from the bio-inspired contribution or the encoder model backbone, in particular since questions on this had not been answered by the authors. Additional doubts were on the sufficient experimental validation relevant to Embodied AI. For all these reasons, the AC judges that the paper is not yet ready for publication.

**Additional Comments On Reviewer Discussion:**

The reviewers engaged with the authors, and discussed the paper with the AC.

---

### Decision · Program_Chairs · 2025-01-22

Reject